# Signaling models for dopamine-dependent temporal contiguity in striatal synaptic plasticity

**Hidetoshi Urakubo**[1¤]*, **Sho Yagishita**[2,3], **Haruo Kasai**[2,3], **Shin Ishii**[1,3]

**1** Integrated Systems Biology Laboratory, Department of Systems Science, Graduate School of Informatics, Kyoto University, Sakyo-ku, Kyoto, Japan, **2** Laboratory of Structural Physiology, Center for Disease Biology and Integrative Medicine, Faculty of Medicine, University of Tokyo, Bunkyo-ku, Tokyo, Japan, **3** International Research Center for Neurointelligence (WPI-IRCN), University of Tokyo Institutes for Advanced Study (UTIAS), Tokyo, Japan

¤ Current address: Division of Cerebral Circuitry, National Institute for Physiological Sciences, Aichi, Japan
* hurakubo@gmail.com

**Data Availability Statement:** Developed MATLAB code and their SBML-style files are available at the public repository GitHub (https://github.com/urakubo/ModelRP.git).

## Abstract

Animals remember temporal links between their actions and subsequent rewards. We previously discovered a synaptic mechanism underlying such reward learning in D1 receptor (D1R)-expressing spiny projection neurons (D1 SPN) of the striatum. Dopamine (DA) bursts promote dendritic spine enlargement in a time window of only a few seconds after paired pre- and post-synaptic spiking (pre–post pairing), which is termed as reinforcement plasticity (RP). The previous study has also identified underlying signaling pathways; however, it still remains unclear how the signaling dynamics results in RP. In the present study, we first developed a computational model of signaling dynamics of D1 SPNs. The D1 RP model successfully reproduced experimentally observed protein kinase A (PKA) activity, including its critical time window. In this model, adenylate cyclase type 1 (AC1) in the spines/thin dendrites played a pivotal role as a coincidence detector against pre–post pairing and DA burst. In particular, pre–post pairing (Ca$^{2+}$ signal) stimulated AC1 with a delay, and the Ca$^{2+}$-stimulated AC1 was activated by the DA burst for the asymmetric time window. Moreover, the smallness of the spines/thin dendrites is crucial to the short time window for the PKA activity. We then developed a RP model for D2 SPNs, which also predicted the critical time window for RP that depended on the timing of pre–post pairing and phasic DA dip. AC1 worked for the coincidence detector in the D2 RP model as well. We further simulated the signaling pathway leading to Ca$^{2+}$/calmodulin-dependent protein kinase II (CaMKII) activation and clarified the role of the downstream molecules of AC1 as the integrators that turn transient input signals into persistent spine enlargement. Finally, we discuss how such timing windows guide animals' reward learning.

**Funding:** This work was supported partly by the Strategic Research Program for Brain Sciences ("Bioinformatics for Brain Sciences") from the Ministry of Education, Culture, Sports, Science and Technology (MEXT), CREST (JPMJCR1652 to SI and HK) from the Japan Science and Technology Agency (JST), the Brain Mapping by Integrated Neurotechnologies for Disease Studies (Brain/MINDS; JP19dm0207001 to SI) from the Japanese Agency for Medical Research and Development (AMED), the International Research Center for Neurointelligence (WPI-IRCN) at The University of Tokyo Institutes for Advanced Study, and JSPS KAKENHI (17K00404 to HU, 17H06310 to SI, and 26221001 to HK). The funders had no role in study design, data collection and analysis, decision to publish, or preparation of the manuscript.

**Competing interests:** The authors have declared that no competing interests exist.

## Author summary

How do animals associate their actions with subsequent rewards (e.g., food)? We previously found a synaptic basis for this association. Synaptic plasticity in the striatum occurs only if the action signal comes just before dopamine (DA) signal, which is known to encode reward/regret information in the brain. Through developing an intracellular signaling model of this synaptic plasticity, we here elucidated the importance of a specific molecule, adenylate cyclase type 1 (AC1). AC1 works as a coincidence detector to associate temporally proximal action and DA signals. The importance of AC1 was shared by two major types of striatal projection neurons: one encodes positive reward-prediction error (RPE) into volume change of the spines, while the other encodes negative RPE. The signaling model also highlighted the importance of the smallness of the spines/thin dendrites for the coincidence detection, and the role of downstream signaling molecules as signal integrators. Taken together, computational modeling of the intracellular signaling strongly promoted our understanding of the mechanisms of this important synaptic plasticity.

## Introduction

Animals behave in their environments to obtain larger rewards [1]. During such reward learning, a specific brain region—the striatum—plays a central role in associating the situation/behavioral action with reward signals [2,3]. Unexpected reward presentation leads to a phasic dopamine (DA) burst (0.3–1 s, DA burst), whereas unexpected reward omission leads to a phasic DA decrease (0.4–2 s, DA dip). To explore the neural mechanism in decoding such "reward prediction error" (RPE) signals, we previously targeted dopamine D1 receptor (D1R)-expressing spiny projection neurons (D1 SPNs) of the striatum, in particular the nucleus accumbens (NAc) core, and found that the DA burst promotes dendritic spine enlargement within a short (~2 s) time window after paired pre- and post-synaptic spiking (pre–post pairing) [4]. The timing dependence of DA burst explains delayed reinforcement [5], so we have designated this form of spike-timing-dependent plasticity (STDP) as reinforcement plasticity (RP).

In D1 and D2 SPNs, RP occurs in response to the two types of inputs: pre–post pairing and DA signal [4,6]. Pre–post pairing mediates large $Ca^{2+}$ influx through voltage-gated $Ca^{2+}$ channels (VGCCs), $N$-methyl-$D$-aspartic acid receptors (NMDARs), and/or $Ca^{2+}$-permeable $\alpha$-amino-3-hydroxy-5-methyl-4-isoxazolepropionic acid receptors (AMPARs) [4,6–8]. DA signal activates D1Rs or dopamine D2 receptors (D2Rs), $G_{olf}/G_{i/o}$ proteins, and then adenylate cyclases (ACs) [4,6]. The interaction between pre–post pairing and DA signal determines the amplitude of and timing dependence in RP. Studies have shown that type 5 AC (AC5) plays predominant roles in both cAMP production and synaptic plasticity in the striatum [9–11]. Thus, two signaling models based on AC5 have been proposed to explain how those signaling dynamics result in timing dependence. One explores the direct effect of DA signal on NMDARs and VGCCs [12], and the other models DA-delay dependence in $Ca^{2+}$ dynamics through a $Ca^{2+}$ buffer [13]. However, those models predict 100-ms and 20-s time windows, respectively, and it still remains obscure about the mechanisms of the in-between time window of ~2 s, which is required for DA-mediated reward conditioning [14]. Further, a pharmacological experiment showed that RP depends on the other type of AC, $Ca^{2+}$-sensitive type 1 AC (AC1) [4], which is also expressed in the striatum [15].

We thus developed the RP models that are based on AC1 to explore signaling dynamics that shapes the time window. In D1 SPNs, the protein kinase A (PKA) activity encodes the input-timing information in RP [4]. Thus, we first focused on the PKA signaling, and elucidated the role of AC1 as a coincidence detector of pre–post pairing and DA burst. We next developed a D2 RP model, which detected the relative timing of pre–post pairing and phasic DA dip. The detection of phasic DA dip was realized by the molecules specifically expressed in D2 SPNs: D2R and adenosine $A_{2A}$ receptors (A2ARs), as suggested by precedent studies [16,17]. We further simulated the downstream signaling molecules as far as $Ca^{2+}$/calmodulin (CaM)-dependent protein kinase II (CaMKII), and clarified their roles as signal integrators. The association of pre–post pairing and DA signal is important for animals' reward learning, which is formalized as the theory of reinforcement learning. We finally discussed the roles of RP in the reinforcement learning based on our computational study.

## Methods

### Overview of modeling

We built computational models of intracellular signaling in RP. The model targeted D1R and D2R SPNs in the NAc core of the striatum, referred to as the D1 and D2 RP models, respectively (Fig 1A, S1 Fig). We aimed to simulate the temporal dynamics of molecular interactions

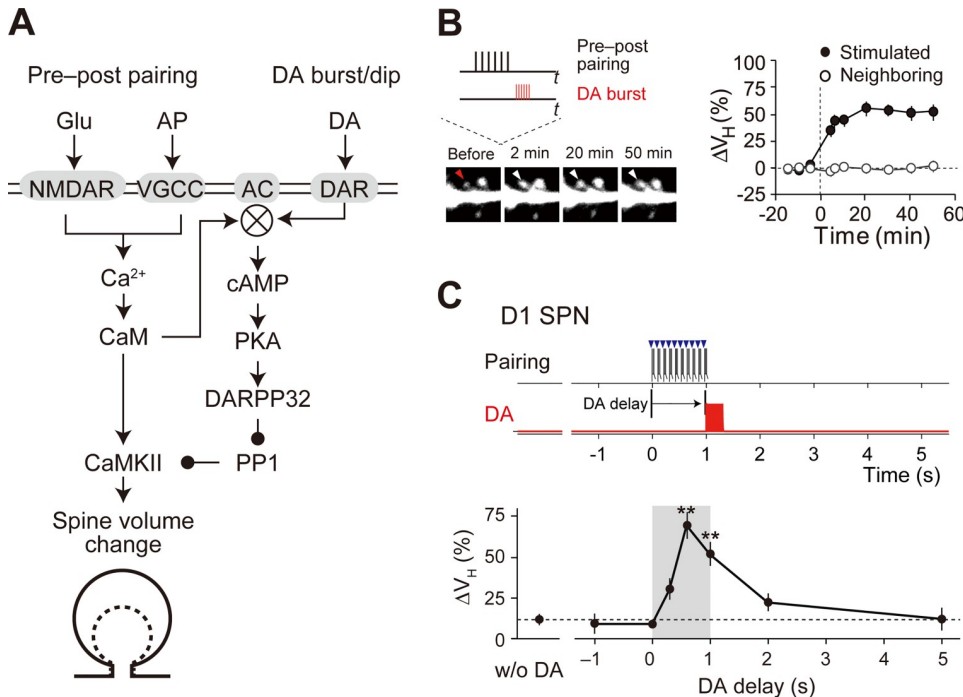

**Fig 1. Reinforcement plasticity (RP) in D1 SPNs.** (A) Signaling mechanisms for RP. The indicated signals were modeled in the present study. Membreane molecules are indicated by the gray shaded areas, whereas the others are cytosolic or extracellular molecules. (B) Enlargement of stimulated spines [4]. Left: time lapse images. The target spine (red and white arrowheads) was stimulated with pre–post pairing (pairing of glutamate uncaging and action potential) and the DA burst with a 0.6-s delay (inset, also see Methods). Right: quantification of the volume change ($\Delta V_H$). From Yagishita et al., Science 26 Sep 2014:Vol. 345, Issue 6204, pp. 1616–1620 (DOI: 10.1126/science.1255514). Reprinted with permission from AAAS. (C) Critical time window for RP [4]. Top and middle: time courses of pre–post pairing (top, blue arrowheads; 10Hz, 10 times) and DA burst (middle, red lines; 30Hz, 10 times). Bottom: increase in spine volume ($\Delta V_H$) against the time delay of DA burst from the onset of pre–post pairing. From Yagishita et al., Science 26 Sep 2014:Vol. 345, Issue 6204, pp. 1616–1620 (DOI:10.1126/science.1255514). Reprinted with permission from AAAS.

as simple as possible (see Binding and enzymatic reactions). Thus, both the D1 and D2 RP models had only a single compartment. Some of parameters in the RP models were based on the preceding models [18–22], and many other parameters were updated according to recent experimental evidence (Tables in S1 Appendix).

The D1 and D2 RP models had two types of inputs: pre–post pairing and DA burst/dip (see Stimulation). The pre–post pairing led to $Ca^{2+}$ signal (Tables B and D in S1 Appendix), and the DA burst/dip led to $G_{olf}/G_{i/o}$ signal (Table C in S1 Appendix). Both of them converged on AC1 (Table E in S1 Appendix). The activated AC1 produced cyclic adenosine monophosphate (cAMP), which led to PKA activation, and the free catalytic (C) subunit of PKA phosphory-lated DA- and cAMP-regulated phosphoprotein 32 kDa (DARPP32) at T34 (Table F in S1 Appendix). Phosphorylated DARPP32 inhibited protein phosphatase 1 (PP1) and thus led to CaMKII activation (Table G in S1 Appendix). In the experiment, the levels of PKA and CaM-KII activities were observed using fluorescence resonance energy transfer (FRET) signals. Those experimental observations were compared with the activities of PKA (free C unit concentration) and CaMKII in the D1 RP model.

Computer simulation was carried out using MATLAB SimBiology (R2018a; MathWorks) under the Suite of Nonlinear and Differential/Algebraic Equation Solvers (SUNDIALS). The developed MATLAB code and its SBML-style files are available at the public repository GitHub (https://github.com/urakubo/ModelRP.git).

## Binding and enzymatic reactions

All molecular interactions in the D1 and D2 RP models were represented by binding and enzy-matic reactions in a deterministic manner [23,24]. For example, a binding reaction in which $A$ binds to $B$ to form $AB$ is expressed by the following equation:

$$A + B \overset{k_f}{\underset{k_b}{\rightleftharpoons}} AB,$$

$$d[AB]/dt = k_f[A][B] - k_b[AB],$$

where $k_f$ and $k_b$ are the rate constants for the forward and backward reactions, respectively. All higher order binding reactions ($> 3$) were decomposed into sets of second-order reactions. This is important because the approximation of higher order binding reactions is inappropri-ate to simulate temporal dynamics of molecules. Enzymatic reactions were modeled based on the Michaelis-Menten formulation:

$$S + E \overset{K_m, k_{cat}}{\longrightarrow} P + E,$$

$$d[P]/dt = k_{cat}[E]/(K_m + [S]),$$

where $S$, $E$ and $P$ denote substrates, enzymes and products, respectively, and $K_m$ and $k_{cat}$ are the Michaelis constant and product turnover rate, respectively, and we did not consider $E$–$S$ complexes for simplicity.

## Stimulation

The D1 and D2 RP models had two type of inputs: pre–post pairing and DA burst/dip. In the experiment, the pre–post pairing constituted of 10-consective elemental pairs at 10 Hz, and each pair comprised a single glutamate uncaging and subsequent 3-consective postsynaptic spiking [4]. We modeled the pre–post pairing by incrementing the level of $Ca^{2+}$ channel

opening ($[CaChannel]$), that represents NMDAR- and VGCC-mediated $Ca^{2+}$ influxes as follows:

$$\{t^i_{\text{pre−post}}\}^{10}_{i=1} = \{0 \text{ s}, 0.1 \text{ s}, 0.2 \text{ s}, \dots, 0.9 \text{ s}\},$$

$$\frac{d}{dt}[CaChannel] = -k_{deact,CaChannel} \cdot [CaChannel] + \sum_i \delta(t - t^i_{\text{pre−post}}),$$

where $k_{deact,CaChannel}$ is the deactivation rate constant of the $Ca^{2+}$ channels, which mainly represents VGCCs (Table B in S1 Appendix), and $\delta$ is the Dirac $\delta$-function. Similarly, DA burst (10 DA spikes at 30 Hz) was modeled by incrementing the current DA level ($[DA]$):

$$\{t^i_{\text{DA}}\}^{10}_{i=1} = \{0 \text{ s}, 0.0333 \text{ s}, 0.0666 \text{ s}, 0.1 \text{ s}, \dots, 0.3 \text{ s}\},$$

$$\frac{d}{dt}[DA] = -k_{dec,\text{DA}} \cdot [DA] + DA_{\max} \cdot \sum_i \delta(t - t^i_{\text{DA}} - t_{\text{DA delay}}),$$

where $k_{dec,\text{DA}}$ is the decrease rate constant (Table C in S1 Appendix), $DA_{\max}$ is the amplitude of the DA signal (Table A in S1 Appendix), and $t_{\text{DA delay}}$ is the DA delay from the pre–post pairing. Finally, DA dip, which is a pause of tonic DA release from many DA fibers, was modeled by

$$\frac{d}{dt}[DA] = -k_{dec,\text{DA}} \cdot \left(DA_{\text{targ}} - [DA]\right),$$

$$DA_{\text{targ}} = \begin{cases} DA_{\text{basal}}, t < t_{\text{DA delay}}, \ t_{\text{DA delay}} + t_{\text{DA dip}} < t \\ DA_{\text{Dip}}, \ t_{\text{DA delay}} < t < t_{\text{DA delay}} + t_{\text{DA dip}} \end{cases},$$

Where $DA_{\text{basal}}$ is the basal DA level (0.5 µM), $DA_{\text{targ}}$ is the target DA level (0 µM), and $t_{\text{DA dip}}$ is the duration of DA dip (0.4 s). All the parameters are described in Tables A and C in S1 Appendix.

## Modeling AC1 activation kinetics

$Ca^{2+}$/CaM is known to stimulate AC1 with two types of time lags: first-order time constant and dead time [25,26], although the mechanism of the dead time is unknown. We thus assumed two intermediate inactive states of an AC1–$Ca^{2+}$/CaM complex ($AC1_{\text{sub1}}$, $AC1_{\text{sub2}}$) as one of the simplest ways to introduce an effective dead time:

$$\text{AC1} + Ca^{2+}/\text{CaM} \underset{xk_{\text{off,AC,CaM}}}{\overset{k_{\text{on,AC,CaM}}}{\rightleftarrows}}$$

$$\underbrace{AC1_{\text{sub1}} \cdot Ca^{2+}/\text{CaM}}_{\text{Inactive}} \underset{k_{\text{down,AC}}}{\overset{k_{\text{up,AC}}}{\rightleftarrows}} \underbrace{AC1_{\text{sub2}} \cdot Ca^{2+}/\text{CaM}}_{\text{Inactive}} \underset{k_{\text{down,AC}}}{\overset{k_{\text{up,AC}}}{\rightleftarrows}} \underbrace{AC1 \cdot Ca^{2+}/\text{CaM}}_{\text{Active}}$$

Where $k_{\text{on,AC,CaM}}$ and $k_{\text{off,AC,CaM}}$ are, respectively, the binding and unbinding rate constants of AC1 to $Ca^{2+}$/CaM, and $k_{\text{up,AC}}$ and $k_{\text{down,AC}}$ are the state transition rate constants, and

$x$ is the allosteric factor (see Table D in S1 Appendix). The active AC1– $Ca^{2+}$/CaM complex showed enzymatic activity only when it further bound to $G_{olf}$ or A2AR. Overall scheme and parameters are described in Table D in S1 Appendix. The inactive states also introduced the first order time constant.

## Modeling domain radius dependence

In the D1 RP model, the spine and soma were considered to be spheres with radiuses of 0.1 μm and 10 μm, respectively [27], which gave the surface-to-volume ratios (SVRs) of $SVR_{spine}$ = ~ 0.3 /μm, and $SVR_{soma}$ = 30 /μm, because $SVR_{spine} = (4\pi r_{spine}^2)/(4\pi r_{spine}^3/3)$. Effective cytosolic concentrations of membrane molecules were determined by the multiplication of the $SVR$ to keep the consistent molecular density per membrane area (Table A in S1 Appendix).

## Fitting simulation to Epac-S$^{H150}$ response

DA-concentration dependence of cAMP levels in the D1 and D2 RP models were fitted to experimentally observed responses of a cAMP sensor, Epac-S$^{H150}$ [16]. Yapo et al. have shown that Epac-S$^{H150}$ did not affect cAMP dynamics [16]. Thus, we simply modeled the Epac-S$^{H150}$ activity, $[Epac]_{bound}$, that obeys cAMP level as:

$$\frac{[Epac]_{bound}}{[Epac]_{total}} = \frac{[cAMP]}{K_d^{Epac} + [cAMP]},$$

where $[Epac]_{total}$ denotes the total Epac concentration, and $K_d^{Epac}$ = 11 μM [28]. Peak $[Epac]_{bound}$ levels were normalized for the fit.

## Fitting exponential decays to the responses of AKAR and Camui

Experimentally observed time courses of AKAR and Camui activities were fitted with exponential decay functions. One thousand least-square fits were conducted against bootstrap replicates of 50 (AKAR) or 43 (Camui) observations, and the decay rate constants (τ) of the 1000 fits were used to obtain the means and 95% confidence intervals (CIs).

# Results

## Requirement of AC1 in the D1 RP model

In Yagishita et al. (2014) [4], two kinds of stimuli were presented to D1 SPNs: glutamate uncaging paired with current injection that elicits post-synaptic spiking (pre–post pairing), and laser irradiation of channelrhodopsin-2 (ChR2)-expressing DA fibers to achieve pre-synaptic DA release (DA burst) (Fig 1B). The pre–post pairing alone did not induce any plastic changes of spines (without (w/o) DA, Fig 1C), but spine volume and synaptic efficacy were both increased when the pre–post pairing further coincided with the DA bursts within a narrow (~2 s) and asymmetric time window (Fig 1B and 1C). In the same study, we also explored the underlying signaling mechanisms in RP (Fig 1A); we found that the PKA signal codes the input timing-dependence (Fig 2D, left), as it shows high correlation with spine enlargement (Spearman's correlation coefficient: 0.94; Fig 2B). Further, the PKA inhibitor PKI completely inhibited the spine volume change [4]. Thus, not only does PKA hold the timing information of the two types of inputs, but also that information is necessary for the expression of RP [13].

The coincidence detection for RP should be realized by an upstream of PKA, presumably ACs [4,29]. Literature showed that AC5 works in the striatum [9,15], and accordingly it has been involved in many computational models [19–21,30]. In our experiments, however, an AC1 inhibitor NB001 blocked the PKA activity of the spines [4,31]. AC1 is known to be

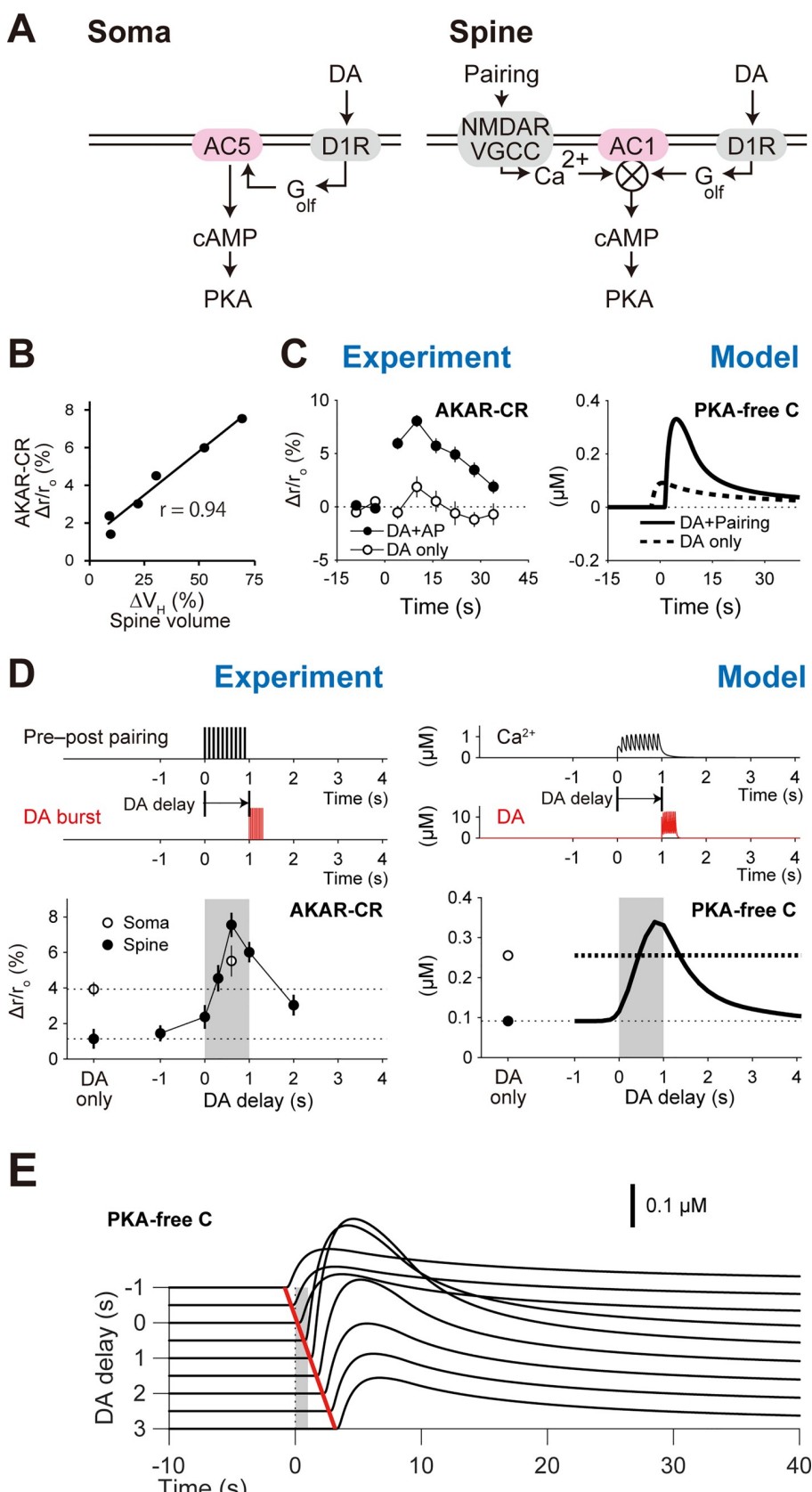

**Fig 2. Requirements of AC1 coincidence detection in the D1 RP model.** (A) Schematics of PKA signaling in the somas and spines. The symbol $\otimes$ denotes the product of two inputs. (B) Spine PKA activity (AKAR2-CR) highly correlates with spine enlargement (r = 0.94, Spearman's correlation coefficient) [4]. From Yagishita et al., Science 26 Sep 2014:Vol. 345, Issue 6204, pp. 1616–1620 (DOI:10.1126/science.1255514). Reprinted with permission from AAAS. (C) Time courses of PKA activity in the experiment (left; DA delay: 0.6 s) [4] and model (right; DA delay: 1 s). From Yagishita et al., Science 26 Sep 2014:Vol. 345, Issue 6204, pp. 1616–1620 (DOI:10.1126/science.1255514). Reprinted with permission from AAAS. (D) Critical time windows for PKA activity in the experiment and model (left and right, respectively). The peak amplitudes of active AKAR2-CR (left) and PKA-free catalytic subunits (right) were plotted. (E) DA-delay dependence of PKA activity. Gray shaded area denotes the periods of pre–post pairing, and red shaded area indicates the periods of DA bursts.

activated only when it binds to $G_{olf}$ and $Ca^{2+}$/CaM simultaneously [32,33]. The $G_{olf}$ signal can be caused by DA burst, and the $Ca^{2+}$/CaM signal is mediated by pre–post pairing. The interaction of those two signals was expected to produce the critical time window for the PKA involvement in RP. We thus built a D1 RP model based on the kinetics of AC1 (Fig 2A).

In the D1 RP model, we simulated the dynamics of molecular interactions based on mass assumption (Figs 1C and 2A, right; Methods). DA burst caused D1R activation, and the activated D1R facilitated GDP–GTP exchange on $G_{olf}$ proteins, followed by the binding of the resultant free $G_{olf}$ to AC1. On the other hand, pre–post pairing caused VGCC/NMDAR-mediated $Ca^{2+}$ influx. The $Ca^{2+}$ was bound to CaM, and $Ca^{2+}$/CaM was further bound to AC1. $G_{olf}$ and $Ca^{2+}$/CaM synergistically activated AC1, and the activated AC1 produced cAMP, leading to PKA activation. As expected, the combination of pre–post pairing and DA burst led to PKA activation, consistent with the experiments (Fig 2C), and the delay of the DA burst reproduced the critical time window for the PKA activity (Fig 2D and 2E) [4]. Thus, in the D1 RP model, AC1 indeed worked for the coincidence detection in RP.

We then examined the D1 RP model that has AC5 but not AC1, because existing studies showed that AC5 dominantly works in the striatum [9,11,15]. In this model, AC5 was activated by DA burst alone (Fig 2D, right; open circle), but additional pre–post pairing did not further amplify AC5 activation, let alone PKA activation (Fig 2D, right; thick dotted line). This result was consistent with the PKA activity in the somas of our experiment (Fig 2D, left; open circles [4]) and other experiments [16,34]. It is thus plausible that AC5 works in the somas whereas AC1 works in the spines and thin dendrites (Fig 2A).

## Temporal contiguity detection by AC1

D1 RP has the asymmetric time window for the DA burst (Fig 2D), which is suitable for detection of a delayed reinforcer. We then addressed how the asymmetry of the time window is shaped (Fig 3A). In the D1 RP model, pre–post pairing resulted in the increase in $Ca^{2+}$ level (Fig 3A, left) [35]. The $Ca^{2+}$ bound to CaM, and the $Ca^{2+}$/CaM stimulated AC1. A precedent study has discovered the involvement of two types of time lags in this process, i.e., first-order time constant and dead time [25,36]. Here, the first-order time constant represents the primary delay of AC1 response against $Ca^{2+}$ signal, where the AC1 response can be fitted by an exponential function (Fig 3A, left; red dotted line, left). The first-order time constant also determines the time-scale of AC1 activity after the end of $Ca^{2+}$ signal (Fig 3A, left; red dotted line, right). The dead time denotes a latent time before AC1 response, and its mechanism is currently unknown. We thus assumed two intermediate "inactive" states of an AC1–$Ca^{2+}$/CaM complex to introduce an effective dead time (see Methods; Table D in S1 Appendix). Note that those inactive states also introduce the first-order time constant. The related model parameters were set to give the dead time and first-order time constant of ~0.3 s (Fig 3A, left, periods between blue dotted lines) and ~2 s, respectively (Fig 3A, right, red dotted lines). On the other hand, DA burst led to the rapid binding and subsequent release of $G_{olf}$ to AC1 (Fig

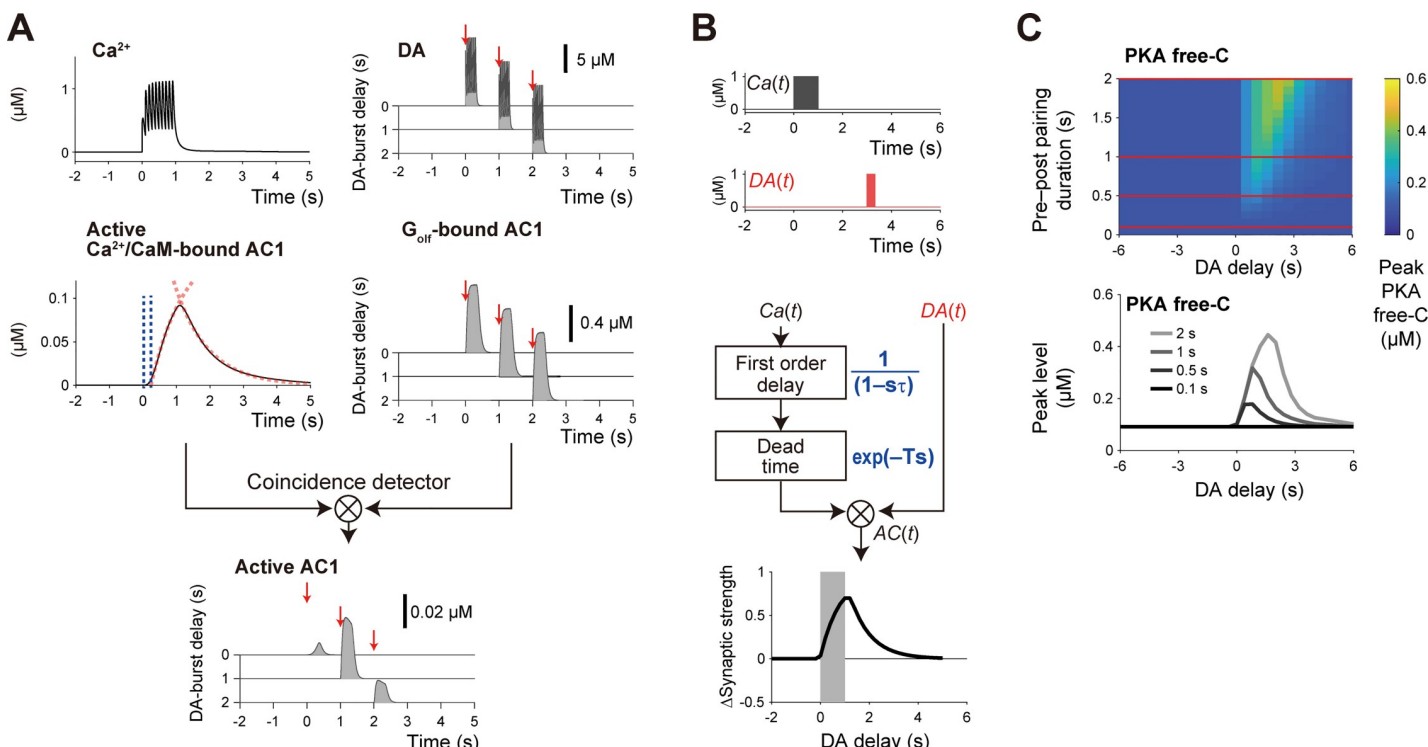

**Fig 3. Delayed activation of AC1 for temporal contiguity detection.** (A) Pre–post pairing stimulated AC1 through $Ca^{2+}$/CaM signal with two types of time lags, i.e., dead time (period between blue dotted lines) and first-order time constant (red dotted lines), and $Ca^{2+}$-stimulated AC1 was activated only during DA burst through $G_{olf}$ binding (right). DA bursts at particular times are plotted (red arrows, 0 s, 1 s, or 2 s). (B) Time window in a simple control model of AC1. $Ca^{2+}$ signal [$Ca(t) = 1$ ($0 < t \le 1$), 0 (otherwise)] was first passed through a first-order time constant [$1/(1+\tau s)$] then a dead time [$\exp(-Ts)$] before AC1 was stimulated. DA signal [$DA(t) = 1$ ($t_{delay} < t \le t_{delay} + 0.3$), 0 (otherwise)] directly stimulated AC1. Here, $T$ is the dead time (0.3 s), $\tau$ is the first-order time constant (2 s), and $t_{delay}$ is the DA delay. The time integration of $AC(t)$ corresponds to PKA Free-C or $\Delta$Synaptic strength. (C) Pre–post pairing ($Ca^{2+}$-signal) duration dependence of the peak amplitudes of PKA-free C in the detailed D1 RP model. The peak PKA free-C values at the red horizontal lines are shown in the bottom panel.

3A, right). Taken together, rapid $G_{olf}$ binding to AC1 occurs during the slow and delayed time couse of the active AC1–$Ca^{2+}$/CaM complex formation, resulting in AC1 activation (Fig 3A, bottom). This shaped an asymmetric time window for the temporal contiguity detection of pre–post pairing and DA signal (Fig 2D). Here, the first-order time constant extended the time window for DA signal by ~2 s beyond the end of pre–post pairing, whereas the dead time ensured no RP by the simultaneous stimulation of pre–post pairing ($Ca^{2+}$ signal) and DA signal.

While the detailed D1 RP model clarified that molecular dynamics can explain the time window, its complexity makes it difficult to understand the roles of AC1 intuitively. We thus built a simple control model (Fig 3B). In Laplace space, a first-order time constant is represented by $1/(1-\tau s)$, and a dead time is represented by $\exp(-Ts)$, where $s$ is the Laplace variable, $T$ is the dead time, and $\tau$ is the time constant. Using these equations, the response of AC1 can be described by:

$$AC\left(t, t_{delay}\right) = L^{-1}\left[\frac{Ca(s)\exp(-Ts)}{1 + \tau s}\right](t) \cdot DA\left(t, t_{delay}\right),$$

Where $L^{-1}[\,](t)$ is the inverse Laplacian, $Ca(s)$ is the $Ca^{2+}$ signal in Laplace space, $DA(t, t_{delay})$ is the DA signal in time space, and $t_{delay}$ is the DA delay. If the $Ca^{2+}$ signal is represented by

a square wave (Fig 3B, top), the AC1 response can be solved as:

$$AC\left(t, t_{\text{delay}}\right) = \left[H(t - T)\left\{1 - \exp\left(-\frac{t - T}{\tau}\right)\right\} - H(t - T - d)\left\{1 - \exp\left(-\frac{t - T - d}{\tau}\right)\right\}\right]$$
$$\cdot DA\left(t, t_{\text{delay}}\right),$$

where $H(t)$ is the Heaviside step function, and $d$ is the duration of the square-wave $Ca^{2+}$ signal. The final readout PKA Free-C or ΔSynaptic strength was defined by:

$$\Delta\text{Synpatic strength}(t_{\text{delay}}) = \int_{-\infty}^{\infty} AC(t, t_{\text{delay}})dt.$$

When the square waves of $Ca^{2+}$ and DA signals were given to the simple model (Fig 3B, top), it produced the time window that was nearly the same as that in the detailed D1 RP model (Fig 3B, bottom). Therefore, the simple D1 RP model successfully captured the mechanism of the temporal contiguity detection.

Because the time window appeared along DA delay (Fig 3B, bottom), one may think that the time lags in the DA signal also lead to the similar asymmetric time window. However, in reality, the DA-signal time lags move the time window in the negative direction. For example, if the dead time in DA signal, $T_{\text{DA}}$, is introduced, ΔSynaptic strength($t_{\text{delay}}$) is changed to ΔSynaptic strength($t_{\text{delay}}+T_{\text{DA}}$), i.e., ΔSynaptic strength(0) appears at $t_{\text{delay}} = -T_{\text{DA}}$. In RP, the longer time lags in the $Ca^{2+}$ signal is essential for the detection of delayed DA signal.

Then, going back to the detailed RP model, we varied the duration of pre–post pairing, and the prolonged time window for the PKA activity was observed when the duration was extended (Fig 3C). This was caused by the progressive increase in $Ca^{2+}$/CaM-bound AC1, which was not saturated at the 1-s pre–post pairing (Fig 3A, left). The simultaneous stimulation of the two inputs did not lead to PKA activation regardless of the duration of the pre–post pairing. Varieties of durations of pre–post spiking are observed in behavioral experiments (e.g., Isomura et al. [37]). Even in such *in vivo* conditions, the D1 RP model robustly detected the temporal order of pre–post pairing and DA signal, suggesting the central role of RP in the temporal contiguity detection in classical conditioning.

## Sensitivity analysis

The detailed D1 RP model was built on many parameters (number of kinetic constants, 58; number of molecular species, 15), some of which may regulate the time window in addition to the AC1 coincidence detection. We thus conducted a sensitivity analysis to clarify what parameters contributed to forming the PKA time window (Fig 4). Here we defined three characteristics of the time window, i.e., the peak amplitude, peak delay, and the full widths at half maximum (FWHM) (Fig 4A), and top twelve sensitive parameters were checked if each of the parameters was changed by ±10% of its original value (Fig 4B and S3 Fig).

In the sensitivity analysis, the amplitude of the time window was mainly affected by the parameters related to $Ca^{2+}$ signal (Fig 4B, left; parameter names colored blue). For example, $k_{\text{influx, CaChannel}}$ denotes the absolute rate of $Ca^{2+}$ influx through NMDARs/VGCCs. This is because the 1-s $Ca^{2+}$ signal was not sufficient to saturate PKA activity, and the increase in the amplitudes of the $Ca^{2+}$ signal further increased the peak PKA activity. Next, the delay of the peak time was mostly governed by the parameter, $k_{\text{down, AC1}}$, which represents a transition rate between the states of AC1–$Ca^{2+}$/CaM complex (Fig 4B, center; see Methods). This indicates that the time lags for AC1 activation was primarily important for the delay of the time window. The FWHM was also governed primarily by the parameter, $k_{\text{down, AC1}}$, in addition to DA-

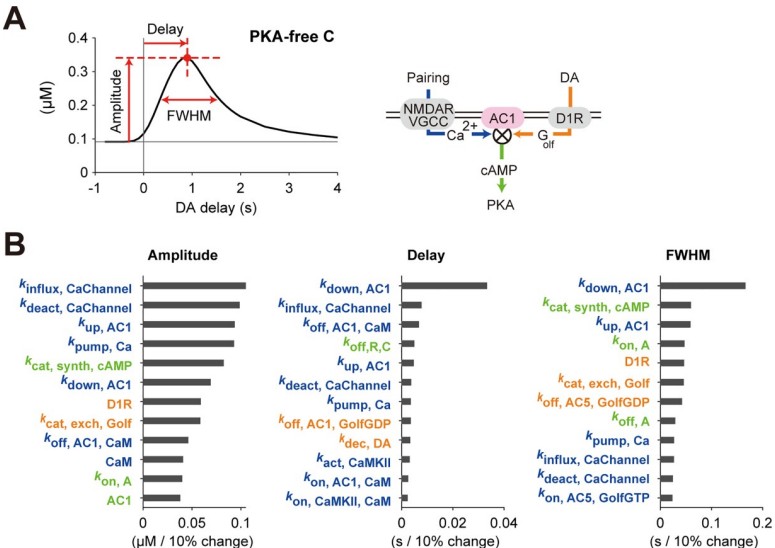

**Fig 4. Sensitivity analysis of the D1 RP model.** (A) Target characteristics of the parameter sensitivity analysis. The changes of amplitude, delay, and FWHM of the time window were quantified against ±10% changes of each of the model parameters, i.e., kinetic constants and molecular concentrations. (B) Top twelve parameters sensitive to the amplitude, delay and FWHM. Parameters regarding $Ca^{2+}$, DA, and PKA signals were denoted by blue, orange, and green, respectively. When a model parameter was changed by −10% and +10%, the corresponding changes of a target characteristic were obtained as $\alpha_1$ and $\alpha_2$, respectively, and the average of their absolute values, i.e., $(|\alpha_1|+|\alpha_2|)/2$ was plotted. Sensitivities against all parameters were shown in S2 Fig.

related parameters (Fig 4B, right; parameter names colored orange). This is because the FWHM was determined based on the convolution of $Ca^{2+}$ and DA signals, and the DA signal is also important as the $Ca^{2+}$ signal. Together, the sensitivity analysis clarified the importance of AC1 delay and validated the effectiveness of the simple model.

## Domain-size dependence of PKA activity

In Yagishita et al. (2014), PKA activity in the somas persisted for longer periods than in the spines or thin dendrites (Fig 5B, filled circles). This persistence was insensitive to a phosphodiesterases (PDE) inhibitor, papaverine (Fig 5B, bottom; red crosses). This observation is explained with the detailed RP model as follows. ACs and phosphodiesterases (PDE) are membrane proteins, and PKA and cAMP are cytosolic molecules (Fig 4A). If a domain is as small as the spines, ACs and PDE in the membranes can rapidly produce and degrade cytosolic cAMP (Fig 5A, top), while they can only slowly produce or degrade cAMP if the domain is as large as the somas (Fig 4A, bottom). More precisely, if we can assume that ACs and PDE are uniformly distributed in the membrane, the number of membrane proteins per domain volume, which is proportional to the SVR, of the somas ($SVR_{spine}$ = ~ 0.3 /μm where radius $r_{spine}$ = ~0.1 μm; Methods) is 100-times smaller than that of the spines ($SVR_{soma}$ 30 /μm where $r_{spine}$ = ~10 μm; Methods). Indeed, when we introduced the small SVR in the somas, the simulated PKA activity decreased much slower than in the spines (Fig 5C, solid lines). The PKA decrease was also insensitive to the PDE blockage (80% decrease in PDE concentration), because of the smaller contribution of PDE in the somas (Fig 5C, bottom; red dotted lines). Thus, the D1 RP model successfully recapitulated the characteristics of PKA activities in the somas and spines by taking their SVRs into account.

We also examined the dependence of the time window for PKA activity on the domain size. A spherical domain was modeled with the uniform densities of membrane AC1 as well as

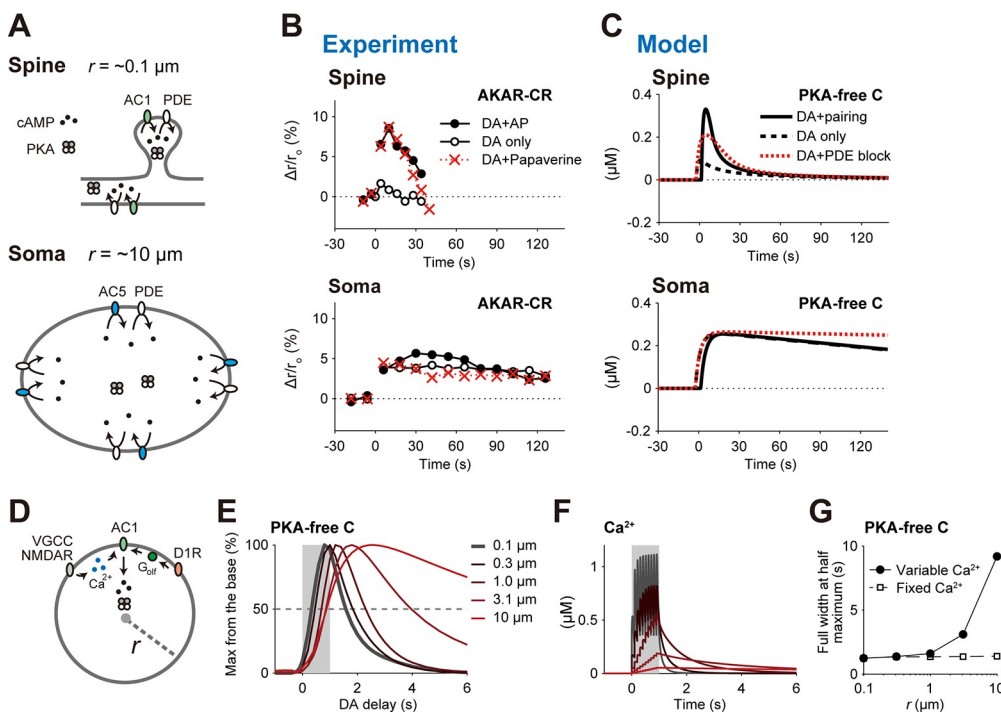

**Fig 5. Smallness of spines was important for short-term PKA activity and its time window.** (A) Domain size affects cAMP dynamics. Spines/thin dendrites are small (radius $r$ = ~0.1 μm), thus having small cytosolic volumes per membrane area (top). The membrane AC1 activity rapidly increases cytosolic cAMP level (the number density of black points), which is rapidly decreased by the membrane PDE. In the somas, cAMP level is increased by the membrane AC5, and slowly decreased by the membrane PDE (bottom). The cAMP levels (the number density of black points per cytosolic volume) are detected by PKA. (B) PKA activities (AKAR2-CR) in the spines and somas in the experiment [4]. Here, DA denotes DA burst, AP designates postsynaptic burst, and papaverine is a PDE inhibitor. The data were taken from the experiment by Yagishita et al. [4]. (C) The simulated PKA activity dynamics with (red dotted lines) and without (black lines) an 80% decrease in PDE. (D–G) Small domain size is necessary for the short time window for PKA activity. (D) A spherical domain with the radius ($r$). In this domain, pre–post pairing activated VGCCs and NMDARs, increasing cytosolic $Ca^{2+}$ level for AC1 activation, while DA burst activated AC1 via membrane $G_{olf}$. (E) Time windows for the PKA activity in the domain with the indicated radiuses $r$. (F) $Ca^{2+}$ dynamics with the radiuses $r$ indicated in (E). (G) Radius dependence of FWHMs of the time windows. The FWHMs under a fixed $Ca^{2+}$ dynamics (same in the case of $r$ = 0.1 μm) were also plotted (dashed line).

other membrane molecules (Fig 5D; Methods) in order to quantify the radius dependence of the time window. The FWHM of the time window was small (~1 s) when the radius $r$ was less than 1 μm, comparable to those of the dendrites ($r$ = 0.1~1.0 μm) or spines ($r$ = ~0.1 μm) (Fig 5E and 5G). However, the FWHMs of the time window became large (> 2 s) if the domain radius was larger than 1 μm (Fig 5E and 5G), like those of the somas ($r$ = ~10 μm). The broad time window was caused by the temporal broadening of cytosolic $Ca^{2+}$ signal, an upstream signal of AC1 (Fig 5F), as in the case of cytosolic cAMP (Fig 5B and 5C). The FWHMs of the time windows remained unchanged when the $Ca^{2+}$ dynamics of the domains was kept identical to that of the spines (Fig 5G, open squares). Together, the domain size critically affected the PKA activity through the interaction of cytosolic and membrane molecules, and the smallness of the spine and thin dendrites was important for shaping the short time window.

## DA-dip detection in the D2 RP model

While D1 SPNs detect phasic DA burst for RP, D2 SPNs detect transient DA dip (0.4–3 s) [16,29,38–40]. We thus built a D2 RP model to examine its DA-dip detectability (Fig 6). The

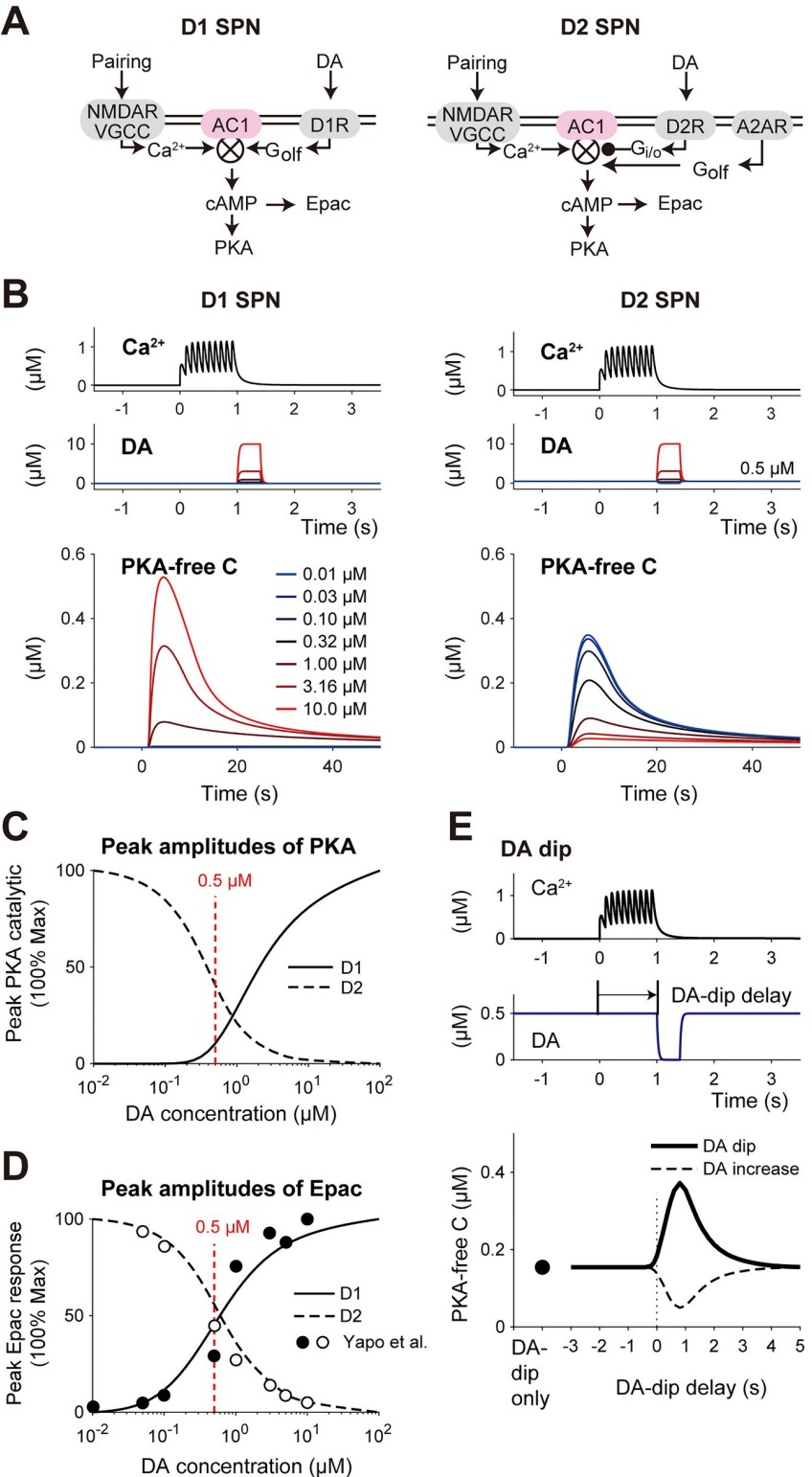

**Fig 6. RP in the D2 RP model.** (A) Schematics of signaling pathways in the D1 and D2 RP models (left and right, respectively). In D2 SPNs (right), AC1 is inhibited by D2R via $G_{i/o}$, and activated by A2AR and VGCC via $G_{olf}$ and $Ca^2$$^+$, respectively. They interact multiplicatively, resulting in PKA activity [45]. (B) PKA responses of the D1 and D2 RP models against pre–post pairing (1-s duration) together with 1-s delayed DA signal (0.4-s duration) with the indicated concentrations. (C) DA concentration dependence of PKA response in the D1 and D2 RP models. (D) The curves

show DA concentration dependence of Epac response in the D1 and D2 RP models (Methods). The overlaid data points were taken from the experiments by Yapo et al. [16]. (E) Time window for PKA activity in the D2 RP model. Thick line: 0-µM DA dip (0.4-s duration). Dashed line: 2-µM DA increase (0.4-s duration).

D2 RP model did not have D1R, but D2R and A2ARs (Fig 6A, right) [41,42]. A2AR is selectively expressed in D2 SPNs in the striatum [6,43,44] (Fig 6A, right), and A2AR is continuously activated by extracellular adenosine, counteracting with D2R [17]. Details of the D2 RP model are described in Tables A–G of S1 Appendix.

The D1 and D2 RP models were used to simulate pre–post pairing (1-s duration) together with DA signal (0.4-s duration) lagging by 1 s, and the PKA responses were quantified (Fig 6B). To check the DA-dip detectability, we introduced 0.5 µM basal DA signal only to the D2 RP model (Fig 6B, right; see Discussion) [16]. In the D1 RP model, the larger DA amplitude led to the larger PKA signal (Figs 6B, left, and 4C), because D1R stimulates AC1 via $G_{olf}$ (Fig 6A). In the D2 RP model, the larger DA amplitude conversely resulted in PKA suppression, whereas the transient DA dip activated PKA (Figs 6B, right and 6C). This is because the DA-dip decreased D2R-mediated $G_{i/o}$ activity, resulting in the relief from the $G_{i/o}$ inhibition of AC1, and the disinhibited AC1 was activated by the stimulation of $Ca^{2+}$/CaM and A2AR-mediated $G_{olf}$ signals (Fig 6A). Note that $G_{i/o}$, $G_{olf}$, and $Ca^{2+}$/CaM multiplicatively regulate AC1 [45]. The DA dose responses of peak cAMP levels in the models showed good fits with those in the precedent experiment using a cAMP biosensor Epac-S$^{H150}$ (Fig 6A and 6D; Methods) [16]. This clear fitting supported the opposite PKA responses between the D1 and D2 RP models.

Finally, we found that the D2 RP model detected the time lag between the pre–post pairing (1-s duration) and the DA dip (0 µM DA, 0.4-s duration), forming a time window for the PKA activation (Fig 6E). The time window was temporally asymmetric as in the case of the D1 RP model, suitable for detecting the temporal order of the two signals. If the DA-dip was replaced with 0.4-s DA increase, the PKA activation was conversely suppressed (Fig 6E, dotted line), showing the possible bidirectional regulation by DA bursts/dip [6,46]. Thus, the D2 RP model, which satisfied a variety of experimental constraints (see Tables A–G in S1 Appendix), successfully detected phasic DA-dip in an input-timing dependent manner.

## Determinants of the time window in the D2 RP model

We also clarified the signaling dynamics transduced to AC1 in the D2 RP model (Fig 7A). As in the case of the D1 RP model, pre–post pairing resulted in the increase of $Ca^{2+}$ level [35] (Fig 7A, top, left), leading to AC1 activation with a dead time and first-order time constant (Fig 7A, middle, left; Table D in S1 Appendix; see Methods). On the other hand, DA-dip signal resulted in rapid unbinding of $G_{i/o}$ from AC1, which in turn activated AC1, due to the continuous $G_{olf}$ signal by A2AR (Figs 6A, right, and 7A, right). The time course of the AC1 activation was thus governed by the $G_{i/o}$-AC1 unbinding process, such as the GTP hydrolysis on $G_{i/o}$ and the $G_{i/o}$-GDP unbinding from AC1 (see Discussion). The time window was asymmetrical with the DA dip lasting for 0.4 s (Fig 6E); the asymmetry allows detection of the temporal order of pre–post pairing and DA dip. We then built a simple model of D2 RP (Fig 7B). The square wave of $Ca^{2+}$ signal was delayed with the two types of time lags (Fig 7B, middle, left) as in the case of the simple D1 model (Fig 3B), and DA-dip disinhibited AC1 due to the relief from D2R-activated $G_{i/o}$, which was represented by a first-order reaction. The total response of AC1 was formalized by:

$$AC_{D2}(t) = L^{-1}\left[\frac{Ca(s)\exp(-Ts)}{1+\tau s}\right](t) \cdot \frac{1}{1+\frac{DA(t)}{K}},$$

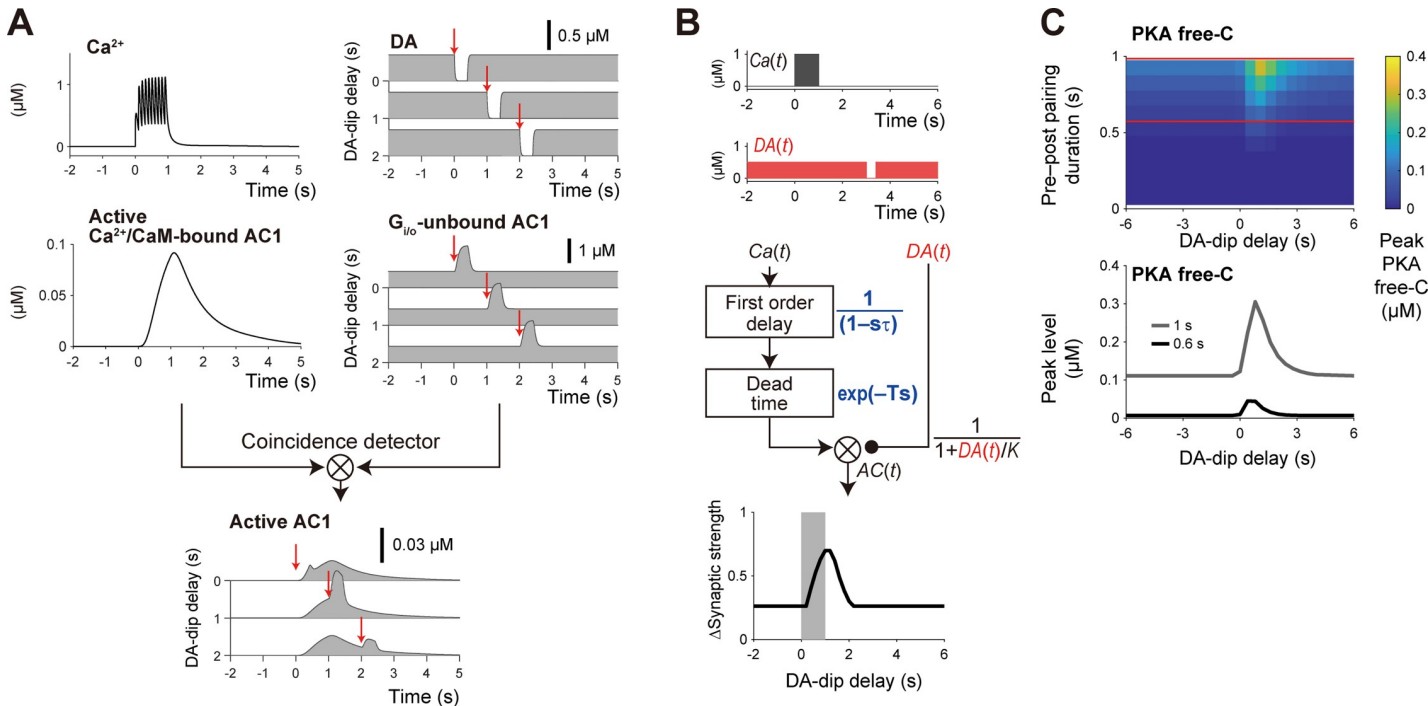

**Fig 7. Temporal contiguity detection in the D2 RP model.** (A) Time lag between $Ca^{2+}$ signal (pre–post pairing) and DA-dip (delay: −1 s, 1 s, or 2 s) was also detected by the AC1 coincidence detector in the D2 RP model. DA dips triggered at red arrow times (top, right) led to relief of AC1 from the $G_{i/o}$ inhibition, and the relieved AC1 was activated by $Ca^{2+}$ burst (left). The rapid unbinding of $G_{i/o}$ activated the active $Ca^{2+}$/CaM-bound AC1, resulting in the asymmetric time window (bottom). (B) A simple model of D2 RP. $Ca^{2+}$ signal $[Ca(t) = 1 \ (0 < t < 1), \ 0 \ (\text{otherwise})]$ was simulated as in the case of D1 SPNs (see Fig 3A), whereas DA dips $[DA(t) = 0 \ (\Delta t < t < \Delta t + 0.4), \ 0.5 \ (\text{otherwise})]$ disinhibited AC1, which was formulated by $1 / (1+ DA(t)/K)$. Here, $T$ was 0.3 s, $\tau$ was 2 s, and $K$ is the effective affinity of DA with D2R (0.3 μM). (C) Pairing duration dependence of the peak amplitudes of PKA-free C in the detailed D2 RP model. The peak PKA free-C values at the red horizontal lines are shown in the bottom panel.

where $K$ is the effective affinity of DA with D2R (0.3 μM). The final readout PKA Free-C or ΔSynaptic strength was the time integration of $AC_{D2}(t)$. The square-wave $Ca^{2+}$ and DA signals (Fig 7B, top) produced the time window (Fig 7B, bottom) similar to that in the detailed D2 RP model (Fig 6E). Thus, the simple D2 model successfully recapitulated the characteristics of the detailed D2 RP model.

We also examined the duration dependence of pre–post pairing in the detailed D2 RP model (Fig 7C). Even when the duration of pre–post pairing was extended, the D2 RP model detects the temporal order between the pairing and DA-dip (Fig 7C). It should be noted that the basal level of PKA activation increased by the longer pre–post pairing, because continuous (tonic) baseline DA signal incompletely inhibited AC1 through $G_{i/o}$ (Fig 7A, right) [6,46], and because the partially-uninhibited AC1 was activated by conjunctive stimulation of A2AR and $Ca^{2+}$ signals, regardless of DA-dip delay. This PKA activity depended on the duration of pre–post pairing or $Ca^{2+}$ signal, which resulted in DA-dip independent baseline PKA signal (Fig 7C, bottom). This suggests the requirement of unknown compensation or adaptation mechanisms for D2 SPNs (see Discussion).

## Downstream signals of AC1 as signal integrators

We then examined the downstream signaling of AC1 as far as CaMKII to clarify their roles in RP (Fig 8A). In the D1 RP model, two input signals synergistically activated AC1. The AC1 produced cAMP, and thus activated PKA. The activated PKA phosphorylated DARPP32 at threonine 34 (T34). Subsequently, the phosphorylated DARPP32 bound to PP1 and inhibited

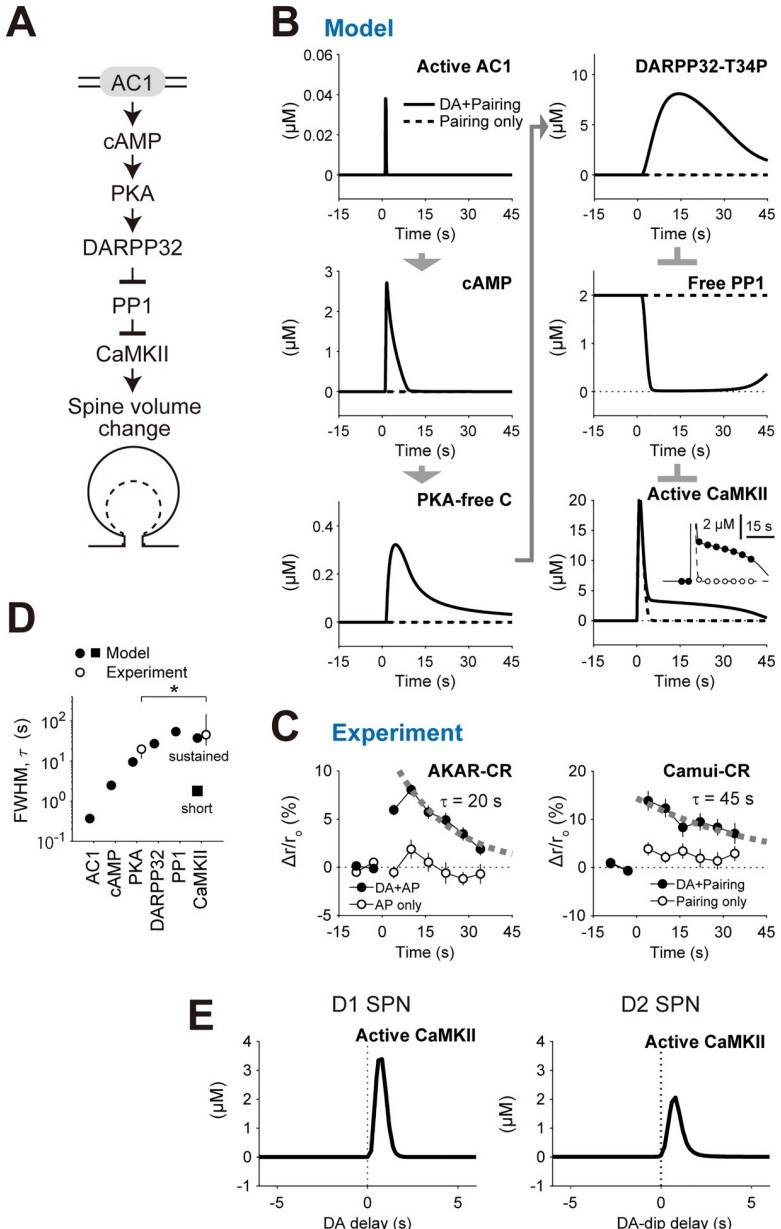

**Fig 8. Downstream signaling of AC1 as temporal integrators.** (A) Signaling downstream of AC1 at the spines. (B) Activities of the indicated molecules. Pre–post pairing together with 1-s delayed DA burst was given to the D1 RP model. Inset in the active CaMKII shows an overlay of the results of discrete sampling of CaMKII activity (filled circles, DA + pairing; open circles, paring only) as used for the experimental results (C, right; 0.2 Hz). (C) Experimentally-observed activities of PKA (AKAR2-CR) and CaMKII (Camui-CR) in D1 SPNs [4]. The AKAR2-CR corresponds to the PKA-free C (B, left, bottom), and the Camui-CR corresponds to the active CaMKII (B, right, bottom). Exponential fits were overlaid. From Yagishita et al., Science 26 Sep 2014:Vol. 345, Issue 6204, pp. 1616–1620 (DOI:10.1126/science.1255514). Reprinted with permission from AAAS. (D) FWHM of the molecular activities in (B) (filled symbols). The CaMKII activity was separated into the short and sustained components (square and circle, respectively). Decay rate constants of the activities of PKA (AKAR2-CR) and CaMKII (Camui-CR) in (C) were overlaid (open circles; mean ± 95% CI, $^{*}p < 0.05$). (E) The downstream molecule CaMKII maintains the time windows for DA delay. The CaMKII activities were taken at 30 s after the stimulation.

its enzyme activity, which led to the activation of CaMKII (Fig 8B). Requirement of this signaling pathway has been validated by pharmacological experiments [4]. The FWHMs of the durations of the AC1 activation, cAMP increase, PKA activation, DARPP32 phosphorylation, PP1 inhibition, and CaMKII activation were progressively increased in the model (Fig 8D, filled circles and square), while the similar difference had been observed in the decay rate constants between AKAR2 (PKA) and Camui (CaMKII) in the experiment (Fig 8C, left and right, respectively; Fig 8D, open circles) [4]. The downstream signaling of AC1 thus worked as signal integrators to transform the 1-s short signal into persistent spine enlargement. Note that the timing information of pre–post pairing and DA burst was successfully maintained even after the signal integration process; The active CaMKII levels 30 s after the stimulation still showed the clear time windows for DA delay both in D1 and D2 RP models (Fig 8E). The CaMKII activation is considered to be tightly connected to the synaptic plasticity [47], although the signaling pathways downstream of CaMKII are still under investigation [48].

## Discussion

D1 SPNs memorize the association between situation/action signal (pre–post pairing) and DA bursts, whereas D2 SPNs retain the association between the situation/action signal and DA dip. This scenario was hypothesized in a modeling study [38] and has been validated experimentally [4,6,39,49]. We here computationally demonstrated that, in both D1 and D2 SPNs, AC1 plays important roles in associating pre–post pairing and such DA signals. The asymmetric time windows support a concept in classical conditioning, the temporal contiguity detection between unconditioned and conditioned stimuli (US and CS, respectively) [50]. The theory of temporal difference (TD) learning predicts the association between situation/action (pre–post pairing) and RPE (DA signals) for reward expectation [51]. Indeed, RP has recently been reported *in vivo* [40,52]. RP should thus play pivotal roles in such reinforcement learning, all of which are important for animals' survival.

In the scheme of reinforcement learning, RP in D1 SPNs should work for solving credit assignment problem, i.e., discovering which choices are responsible for rewards in multiple situation/action signals with multiple delay time [53,54]. Situation/action events activate specific corticostriatal and thalamostriatal fibers, some of which are selectively strengthened if their activation coincides with phasic DA bursts (Fig 1). This strengthening results in the reinforcement of specific situation/action signals that are specifically associated with DA bursts. When situation/action signals are prolonged, such events can be decomposed into a series of phasic signals to the striatum (delay lines), owing to other neuronal systems, e.g., the hippocampus [55,56]. Selective strengthening of some of the delay lines is indeed equivalent to the process of original TD learning [51]. Thus, RP in D1 SPNs is a plausible neural implementation of the TD learning.

In the simple model, we introduced two types of lag times for $Ca^{2+}$-induced AC1 activation, i.e., first-order time constant ($\tau$) and dead time ($T$). The first-order time constant formed the time window for delayed DA signal (Figs 2D and 3A). The dead time was also necessary to ensure no RP by the simultaneous stimulation of $Ca^{2+}$ and DA (Figs 1B and 2D); if DA and $Ca^{2+}$ signals are both given at onset time 0, the DA signal rapidly activates AC1, while the $Ca^{2+}$-induced AC1 activation is postponed during the DA-activation period (Fig 3A). Onyike et al. originally discovered those time lags ($\tau$ = 13 s and $T$ = ~2.5 s) [25], but they are too long for RP in the NAc. Because their experimental environment (artificially expressed AC1 in isolated insect cell membranes) is completely different from SPNs in the NAc, we set these rate constants so as to produce the time window ($\tau$ = 2 s and $T$ = 0.3 s). The present study highlights the importance of identification of the time lags because they are directly related to animals' behaviors in classical conditioning.

In addition to the NAc, RP has been reported in multiple brain regions [54,57,58]. Shindou et al. discovered similar RP in the dorsomedial striatum, and its 2-s time window should be based on the similar mechanism [59]. He et al. also discovered RP in the visual cortex that depends on serotonin (5HT) or norepinephrine (NE) signals [60]. NE binds to β2A receptors, activating $G_s$ (similar to $G_{olf}$). Thus, NE-dependent LTP may also depend on AC1 coincidence detection. On the other hand, 5HT binds to $5HT_{2C}$ receptors, which does not couple $G_{olf}$, $G_s$, or $G_i$, indicating the involvement of other coincidence-detection mechanisms. Also, Brzosko et al. reported hippocampal RP that has a 30 min time window [58]. The long-time window cannot be achieved by AC1 kinetics, but by other mechanisms, such as cAMP response element binding protein (CREB)-regulated gene expression. Thus, multiple mechanisms work for coincidence detection or eligibility trace in RP.

The peak of PKA activity in the D1 RP model appeared at the DA delay of 1 s (Fig 2D), whereas the peak in the experiment appeared at 0.6 s (Figs 1B and 2D). This difference can be resolved by introducing an amplification factor of AC1 activity during pre–post pairing (S3 Fig, top, left). DA fiber stimulation was given for 0.3 s (red arrows in S3 Fig, top, right); thus, $G_{olf}$-bound AC1 also appeared for 0.3 s (curve in S3 Fig, top, right). The $G_{olf}$ binding is a prerequisite for AC1 activation. When it was convoluted with the amplification factor (S3 Fig, top, left), the time window showed the amplification ranging between −0.3 s and 0.7 s. The subsequent multiplication with $Ca^{2+}$ signal resulted in the modified PKA time window with a 0.6-s peak as observed in the experiment (S3 Fig, bottom). Indeed, AC1 is reported to be amplified by membrane depolarization (postsynaptic spiking) through the increase in extracellular $K^+$ level [61]. Further studies are necessary to confirm the cause of the 0.6-s peak more clearly, but this suggests that such modulatory factors also affect DA-delay dependence in RP.

In the D1 RP model, cAMP levels in dendrites were increased by AC1, and those in somas were increased by AC5 (Figs 2D and 5A–5C). Those settings explain the time window for DA delay (Fig 2D) as well as the time course of PKA activation (Fig 5A–5C). However, this does not denote the actual localization of AC1 and AC5 in dendrites and somas, respectively. In the striatum, AC5 gives a ~80% in the total cAMP levels of neurons [9], is expressed not only in the somas but also in the dendrites of neurons [62], and contributes to synaptic plasticity [11]. Considering our studies, AC1 and AC5 should rather be functionally separated. In particular, upon optogenetic stimulation of DA fibers to give phasic DA signals, the short stimulation may give spatiotemporally limited G-protein signaling only for AC1 activation. Indeed, AC1 is known to be sensitive to transient $Ca^{2+}$ signal [26]. Given that AC5 should also play roles in synaptic plasticity, AC5 may contribute to the time window in other forms of RP [13].

The multiple roles of D2 SPNs have been described. One study reports that the spiking of D2 SPNs encodes no-reward outcome and next-action selection [63], and another shows that the artificial activation of D2 SPNs leads to freezing and escape behaviors of mice [64]. Together, D2 SPNs work for the change of animal's current behavior into more favorable one, and this process requires detection of DA-dip as a regret signal [40]. In the present study, the phasic DA dip was successfully detected by the D2 RP model with a set of plausible parameters (see Tables A–G of S1 Appendix). Although we did not address its parameter dependence in the present study, the DA-dip detection was determined not by activation rate constants, but by unbinding and deactivation rate constants of the $G_{i/o}$ signaling. This is because the sudden decrease of DA triggers D2R deactivation and the subsequent $G_{i/o}$ deactivation process. The $G_{i/o}$-GTP hydrolysis rate (deactivation rate) should be set within a certain range, and the deactivated $G_{i/o}$-GDP must be unbound from AC1 rapidly. Such parameter requirements for DA-dip detection will be addressed in the future study.

Several research groups have already proposed signaling models of RP. First, Yarali et al. developed a model of *Drosophila* aversive learning [36]. In this model, AC detects the temporal

order of US/CS-activated molecules, producing a 100-s time window. The wider time window is mainly due to the slower dynamics of G-protein and $Ca^{2+}$ signals, e.g., their activation time constants are 7 s and 10 s, respectively, but further studies are necessary to clarify such species-dependent difference in RP signaling. Nakano et al. have also developed a model of RP [12]. They considered acute DA effects on NMDARs and VGCCs within several hundred milliseconds, and predicted larger $Ca^{2+}$ signals when the DA signal precedes postsynaptic spiking. In our experimental conditions, we did not see any DA signal-dependent change of $Ca^{2+}$ level [4], but such receptor/channel modulation may affect the excitability of SPNs [17]. Nair et al. have also simulated $Ca^{2+}$- and DA-timing-dependent CaMKII activity [13], linking to RP. They showed the time window for CaMKII activation that is caused by input-timing dependent phosphorylation of DARPP32 and a $Ca^{2+}$ buffer. However, their model, in principle, does not show any $Ca^{2+}$ and DA timing dependence in PKA activity, going against the clear dependence shown in our experiments (Fig 2D). Together, only the current model has successfully described RP that is dependent on pre–post pairing and DA burst/dip, taking timing dependence as well as signaling mechanisms into account.

The current D1 and D2 RP models were single-compartment models, and no-spatial factors were considered (Methods). This simplification facilitates conceptual understanding, but it leaves important issues that need to be addressed in future studies: spatial interaction between dendrite-wide PKA signal (Fig 2C) and stimulated synapse-specific spine enlargement (Fig 1B). In Yagishita et al., PKA was activated in a dendrite-wide manner [4], which was presumably the result of dendritic AC1 activation in response to dendritic $Ca^{2+}$ and $G_{olf}$ /$G_{i/o}$ signals. The dendritic $Ca^{2+}$ is mediated by VGCC through postsynaptic spiking, and DA fiber activation gives spatially-nonspecific DA signal [65], activating dendritic D1R/D2R for the dendritic $G_{olf}$ /$G_{i/o}$ signals [66]. On the other hand, focal uncaging of glutamate triggers stimulated synapse-specific CaMKII activation, resulting in specific spine enlargement. The interaction between those two signals may determine the spatial extent of RP in the dendritic space. This issue will be targeted by reaction-diffusion models of intracellular signaling [67].

*In vivo*, DA fibers in mice tonically deliver spikes at ~5 Hz (e.g., Cohen et al. [68]), and the tonic DA firing gives basal DA signal. While measured basal DA concentrations differ depending on literature, ranging from 10 nM to 1 μM [69–71], we set the basal DA concentration at 0.5 μM on the basis of a microdialysis experiment [71] (Fig 4B–4E). Some researchers consider the basal DA concentration to be a few-dozen nanomolar [29,72], which is close to D2R's affinity for DA in the high affinity state of D2R ($K_d$: ~ 40 nM) [73], i.e., in which the D2R is pre-coupled with $G_{i/o}$ [74]. The basal DA is within the dynamic range of the D2R; however, the high affinity state of D2R has slow unbinding rate constant of DA ($t_{1/2}$ = 80 s), unsuitable for detecting 0.4-s DA dip [75]. Rather, the dynamics of striatal DA signal *in vivo* can be observed using genetically-encoded fluorescent DA sensors that have $K_d$s of 0.3–1.6 μM [76], and D1R and D2R SPNs show cAMP responses against sub-micromolar DA signals [16,77] (Fig 4D). Thus, basal DA concentration seems to lie within a sub-micromolar range, and D1R and D2R may operate under their low affinity states ($K_d$ of D1R: ~ 5 μM; $K_d$ of D2R: ~ 2 μM) [73]. Indeed, fitting our models to the DA-dose responses of cAMP (Fig 4) gave the $K_d$s of 2 μM (D1R) and 10 μM (D2R), both of which are rather close to those of their low affinity states. In the low affinity state, D1R and D2R do not form a stable complex with G-protein, but activate many surrounding G-proteins [74]. Although our results support the low affinity states of D1R and D2R, much controversy exists over this topic, and further studies will be necessary to draw the conclusion.

In the present study, we did not simulate the D1 RP model under the basal DA signal. This was because it was difficult to build a quantitative model of the adaptation mechanisms of D1R. Most prominently, while the DA–D1R binding results in PKA activation (Fig 2A), intracellular

PKA activity in turn leads to a 100-fold decrease of D1R's affinity for DA and D1R's maximal activity also decreases by 100-fold with a time constant of 10 min [78]. Thus, the D1R–ACs–PKA signaling constitutes a negative feedback system to limit PKA response solely to phasic DA signal. Similar adaptations also exist in D2R SPNs [79,80], and part of them should be regulated by immediate early genes, which are beyond the scope of the present study. In short, the DA sensitivity of PKA is modifiable both in D1R and D2R SPNs, especially by the basal DA signal itself. Quantitative modeling of such adaptations is an important future direction, because they could be related to Parkinson's disease and drug abuse. Symptoms of Parkinson's disease appear after over 50% loss of DA fibers [81], and the robustness against the DA loss should partly come from the above mentioned adaptations. Chronic cocaine intake increases DA level in the NAc [82] as well as the levels of cAMP and CREB [83], all of which are expected to affect RP. If those factors are successfully taken into consideration, the RP models presented here will become more powerful to give a comprehensive view of the whole DA system for RP.

## Supporting information

**S1 Fig. Overview of the D1 and D2 RP models.** Left: D1 RP model. Pre–post pairing and DA burst/dip were inputs, and PKA activity, CaMKII activity, and synaptic volume change were readouts in the experiment. Right: D2 RP model. Unlike the D1 RP model, DA dip disinhibited AC1 via the relief from $G_{i/o}$ inhibition, while pre–post pairing and A2AR activated AC1.
(TIF)

**S2 Fig. Overall results of the parameter sensitivity analysis on the D1 RP model.** The changes of amplitude (A), delay (B), and FWHM (C) of the time window were quantified when each target kinetics constant or molecular concentration was changed by ±10%. The mean changes of the top twelve changes were plotted in Fig 4.
(TIF)

**S3 Fig. Amplification of AC1 during pre-post pairing resulted in the peak of PKA at a 0.6-s DA delay.** The amplification factor of AC1 activity during pre-post pairing (top, left) was convoluted with DA-dependent $G_{olf}$ stimulation for 0.3 s (top, right). This convolution resulted in the amplification of AC1 between −0.3 s and 0.7 s (middle). The amplification was further multiplied with $Ca^{2+}$ signal, which resulted in a PKA time window with a 0.6-s peak, as observed in the experiment (bottom, blue). Red arrows denote the times of DA-fiber stimulation (top, right). The amplification level was set to be ×1.45 to give a best fit with the experiment.
(TIF)

**S1 Appendix. Detailed description of D1 and D2 RP models.** Detailed reactions, molecular concentrations, and reaction rate constants.
(PDF)

## Acknowledgments

We would like to thank Yoshihisa Fujita and Kouichi C. Nakamura for their helpful comments.

## Author Contributions

**Conceptualization:** Hidetoshi Urakubo, Sho Yagishita, Haruo Kasai, Shin Ishii.

**Investigation:** Hidetoshi Urakubo.

**Resources:** Sho Yagishita.

**Supervision:** Haruo Kasai, Shin Ishii.

**Writing – original draft:** Hidetoshi Urakubo, Sho Yagishita, Haruo Kasai, Shin Ishii.

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
