## [Decision Letter · Decision Letter 0]

8 Nov 2019

Dear Dr Urakubo,

Thank you very much for submitting your manuscript 'Signaling models for dopamine-dependent temporal contiguity in striatal synaptic plasticity' for review by PLOS Computational Biology. Your manuscript has been fully evaluated by the PLOS Computational Biology editorial team and in this case also by independent peer reviewers. The reviewers appreciated the attention to an important problem, but raised some substantial concerns about the manuscript as it currently stands. While your manuscript cannot be accepted in its present form, we are willing to consider a revised version in which the issues raised by the reviewers have been adequately addressed. We cannot, of course, promise publication at that time.

Your revisions should address the specific points made by each reviewer, especially the matter of model predictions raised by reviewer 2. Please return the revised version within the next 90 days. If you anticipate any delay in its return, we ask that you let us know the expected resubmission date by email at ploscompbiol@plos.org. Revised manuscripts received beyond 60 days may require evaluation and peer review similar to that applied to newly submitted manuscripts.

Sincerely,

Joanna Jedrzejewska-Szmek, Ph.D.

Guest Editor

PLOS Computational Biology

Kim Blackwell

Deputy Editor

PLOS Computational Biology

[LINK]

Reviewer's Responses to Questions

**Comments to the Authors:**

Reviewer #1: the review was uploaded as a pdf file

Reviewer #2: In this article, Urakubo et al propose a computer model for the kinetics of the signaling pathways underlying dopamine-mediated reinforcement of the eligibility trace triggered by STDP protocols at the corticostriatal synapse. More specifically, a previous experimental article by the authors (Yagishita et al, Science 2014) suggested that dopamine signaling in the dendrites of the striatum neurons would be based on the Ca-dependent AC1 isoform and on the more classical Ca-independent AC5 isoform in the soma. In the submitted paper they develop a mathematical model of the kinetics of AC1 and AC5 signaling, and explore with the model how signaling gives rise to the experimentally observed critical time window for Dopamine (DA), that is, the observation that to trigger plasticity, DA stimulation has to arise in a ~2 s time window after the electrical pairing (prepost) stimulation.

I certainly appreciate the attention to a very important problem, because, in a way, the article consists in questioning the molecular bases for the resolution of the distal reward problem in the striatum, i.e. the fact that in many reinforcement tasks, the reward comes well after the action that triggered it and is to be rewarded. I also definitely appreciate the use of a modelling approach to tackle this question. However, I must admit I am not sure that, eventually, the new information brought about by the model is enough to justify publication in PLoS Computational Biology

Major points:

1. Overall, what do we learn from the mathematical model in the paper? The presence of AC1 in the dendrites means that both Ca and DA-Golf are needed to trigger PKA-cAMP signaling, so of course, PKA signaling is significant only when the electrical pairing stimulation is superimposed to/coincident with DA stimulation. Since this electrical stimulation lasts for 1 s here (10 pairings @ 10 Hz), the window of efficient DA application is thus expected to be roughly of 1 second after the beginning of the electrical stimulation. In my opinion, this is very much expected and I think there is no need for a mathematical model for that. Since the critical time window of DA application is the major focus of the paper, I am not convinced the new information obtained with the model is enough for publication in PLoS Comput Biol.

2. To be fair, I could accept the argument that the interest here is not only on the duration of the critical time window, but also on the exact shape of the dependence of the signaling amplitude to DA delay. Indeed, it is stressed out by the authors that this dependence forms an asymmetric peak where:

i) there is no PKA signal if both stimulations (electrical and DA) starts simultaneously (zero DA delay).

ii) PKA signaling remains strong even when the electrical stimulation is finished; in fact, experimentally, PKA signaling seems to remain significant even for 2s delays ie when DA arrives more than 1s after the last electrical pairing.

While I agree those are puzzling observations, I am not convinced by the interest of the submitted article regarding those points.

2A) First, in terms of experiments, the dependence of PKA signaling to DA delay (ie Fig 1B in the article) around the DA window is based on too few points to be affirmative of its exact shape. At best, the conclusions of the authors rely on only 5 observed DA delays (0, 0.3, 0.6, 1 and 2s). I think one would need a better sampling of the DA delays in the [0-2s] range to ascertain what is the shape of the dependence.

2B) To explain the above observations, the authors' model proposes that the whole shape of the critical time window relies on the Ca2+-bound AC1 response (Fig 5A) that, in the model, shows a long latency after the beginning of the electrical stimulation (peaking at the end of the stimulation) and a very long decay to basal values after the electrical stimulation is finished (more than 3 second). In effect, the proposal here is that the "eligibility trace" consists in the Ca2+-bound AC1 response. However, this proposal has already been made based on experimental observation of the time course of AC1 activation after Ca activation. For instance, Onyke et al, J Neurochem 2002 (cited by the authors as ref 35) already noted 17 years ago that the AC1 response was delayed after the onset of the Ca stimulation, peaked only at termination of the calcium signal, and decayed back to basal only after several seconds. The significance of this particular temporal shape in the framework of delayed rewards was even clearly understood and discussed in the paper. The modelling results presented in the submitted article are mostly an illustration of this idea. But I am sorry that I do not see exactly what new significant information is produced by the submitted paper compared to e.g. Onyke et al.

3) Note that actually, the model is not perfectly emulating the experimental timing window. Notwithstanding the critics expressed above related to the number of DA delay experimental points, in the experiments, the maximal delay is of 0.6 sec (ie activation at 0.6 s is larger than at 0.3 or 1 s) whereas in the model the peak delay is 1 s (that is why figure 2C uses two DA delays, one for the model and one for the experiments). This discrepancy should be studied in the paper. It might be a sign that all is not perfectly understood in this system.

4) Pushing back the details of the mathematical model to the supplementary information is not suitable for the readership of PLoS Comput Biol. The readers will want to know what exactly is in the model before reading the result section. Likewise modelling-inclined readers looking at Figure 2D will immediately wonder how come that PKA is activated with a 2s DA delay, for which DA arrives at a point where the calcium trace has reached back its baseline for more than 1s. Of course the answer comes in Figure 5, but I think this is too long a delay between the two figures. I strongly advice to relocate current Fig 5 just after Fig2, so as to explain PKA activation with 2s DA delay right after illustrating it.

Minor Points

1) It is not clear what is plotted in Fig5A middle: is it **active** Ca/CaM-bound-AC1 or total Ca-boud-AC1 (since AC1 has several states here).

2) What causes, in the model, the latency of Ca/CaM-bound-AC1? Is it the addition of the two intermediate inactive states for the AC1-Ca/CaM complex or the overall kinetics of Ca binding to CaM (either alone or in complex with AC1). Line 245 says:

"The Ca2+/CaM bound to AC1 with a rate constant of ~1 s, and the Ca2+/CaM-bound AC1 was activated after a latent time of ~0.3 s". That seems to imply that the major part of the latency is due to Ca/CaM binding to AC1 and not to Ca2+/CaM-bound AC1 activation.

3) Yagishita et al, Science 2014 is not the only experimental paper that questioned the critical time window of DA application after STDP in SPNs. Other papers exist, including Fisher et al, Nature Comm, 8:334, 2017 or Brzosko et al, eLife, 4:e09685, 2015 and must be cited. And their results must be discussed in the light of Yagishita et al, Science 2014 and the results of the present submission.

4) I do not understand the bottom panel of Figure 5D: the peak level of PKA Free-C is much lower with 0.6 sec electrical pairing stimulations than with 1 sec ones, even for huge delays, e.g. -6 or + 6 seconds. With such large delays there is no interaction at all between the electrical and the DA stimulations. So there's no reason a 0.6 sec pairings would give such a lower peak of PKA free C with eg +6 or -6 sec DA delays. If it the difference is real and not an error of the plot, it must be explained.

5) lines 468-469: from these equations, it seems that each prepost pairing increases the VGCC rate and the NMDAR rate by exactly 1 unit. I do not understand how the model can work without a parameter to modulate the amplitude of these increments (as is the case for DA rate for instance).

5) Fig 6C: around line 290, the authors claim that the FWHMs of the duration of the CaMKII response is larger than that of the PKA response in Fig 6C. This would agree with their model predictions. Unfortunately, in the absence of any quantification of figure 6C, I must say that this claim is not at all obvious from visual inspection of the figure.

**Have all data underlying the figures and results presented in the manuscript been provided?**

Reviewer #1: Yes

Reviewer #2: No: The manuscript indicates that the Matlab code used to simulate the model is available on a github repository. Unfortunately, the corresponding url does not seem valid, so that it is not possible to check the simulation code (I have found the code for a previous paper on the github of the first author, but not for the submitted paper).

PLOS authors have the option to publish the peer review history of their article (what does this mean?). If published, this will include your full peer review and any attached files.

Reviewer #1: No

Reviewer #2: No

---

## [Decision Letter · Decision Letter 1]

21 Apr 2020

Dear Dr Urakubo,

Thank you very much for submitting your manuscript "Signaling models for dopamine-dependent temporal contiguity in striatal synaptic plasticity" for consideration at PLOS Computational Biology. As with all papers reviewed by the journal, your manuscript was reviewed by members of the editorial board and by several independent reviewers. The reviewers appreciated the attention to an important topic. Based on the reviews, we are likely to accept this manuscript for publication, providing that you modify the manuscript according to the review recommendations.

Sincerely,

Joanna Jędrzejewska-Szmek, Ph.D.

Guest Editor

PLOS Computational Biology

Kim Blackwell

Deputy Editor

PLOS Computational Biology

[LINK]

Reviewer's Responses to Questions

**Comments to the Authors:**

Reviewer #1: The authors have tried to address most of the previous concerns, so I'm happy to approve the publication. This is an important paper as it addresses a very important phenomenon that is important for reward learning.

Reviewer #2: The authors have made significant changes since last review which has improved the manuscript and clarified some of my concerns. However, I believe there remains a few points to be addressed as noted below:

MAJOR POINTS

1) The authors now systematically refer to AC1 stimulation as having two types of time lags (dead time and first-order delay). I understand this necessity because of the equation for the "simple model" shows two constants \\tau and T. But for the reader who is not an expert in dynamics systems, it must be explained what these two time lags correspond to, in terms of dynamics. In particular, it must be explained that the "first-order delay" also sets the time-scale of the decay of the AC1 signal once the calcium signal is off.

Moreover, I believe that the usual term in dynamical systems textbooks is not "first-order delay" but "first-order time constant".

2) Figure 2C: The model for "DA+pairing" shows a dead-time that is similar to the experimental curve. However, the experimental data for "DA only" also exhibits a similar dead-time, a feature that is not reproduced by the model (no dead time in the model for "DA only"). I believe this might be a hint that the hypothesis made by the authors that the dead time originates entirely from AC1 activation by Ca2+/CaM may be flawed. Indeed, if the dead time survives in the absence of Ca2+/CaM stimulation, one may reasonably argue be that the dead-time is not due to Ca2+/CaM activation of AC1. I believe that the current manuscript does not treat this point seriously enough.

3) Simple model for AC1 response:

a) I understand that the equation given line 305 is written for simplicity in terms of its Laplace transform, but I think it would be very useful for the less signal processing-inclined readers to also give the inverse transform of the first term of the RHS member for e.g. a square-pulse calcium, i.e. $\\theta(t-T) (1-\\exp(-(t-T)/\\tau)) - \\theta(t-T-d) (1-\\exp(-(t-T-d)/\\tau))$ where $d$ is the duration of the square-pulse calcium input and $\\theta$ is the Heaviside function.

b) the conventions used in Figure 9 are at odds with the other figures of the paper. That is quite misleading. For instance, in Figure 1-8, the delay between pre-post stimulation/Ca signal and DA application is counted starting from the beginning of the pre-post stimulation/Ca signal whereas the "interval" of figure 9 starts at the end of the pre-post stimulation/Ca signal. As a result, for the NAc data, the plasticity window is [0,4] seconds in Fig 1-8 but [-1,3] seconds in Figure 9. Moreover, the Ca-DA delay is most of the times referred to as "DA delay" (Fig 1,2,4,5,8), but sometimes "\\Delta t" (Fig 3B, 7B), "DA-burst delay" (Fig 3C) or "Interval" (Fig 9). Please homogenize notations.

c) There seems to be potential inconsistencies between the bottom line of figure 9 and the main text. In B, the figure reads '$\\tau=2 s, T=2.14 s$ whereas the main text indicates that the best fits were obtained with $T=5$ and (probably independent fit) $\\tau = 0.6s$. Likewise, for C, the figure says $\\tau=2 s$ whereas the main text instead points to $\\tau = 5.06$. Moreover, I think it would help the reader if Fig 9 would also plot the delayed AC1 signal (i.e. the inverse Laplace transform of the equation line 305), in addition to the Ca and DA pulses/signals of the top line. That would be useful to realize in practice what the estimated parameter values actually mean.

d) While I acknowledge the value of inclusion of the "simplified model", I believe the authors should be more cautious with their interpretation of the fits of Fig 9. First, the number of data points used for the fits is really very very small (4 points for the DMS, only 3 points for the cortex!). Moreover, the values of the estimated parameters, especially the dead-time, are very large (from 2 to 7 s). It is difficult to understand how the simple signaling mechanism of Fig 1C could give rise to such large dead-times. Note that I am not saying that this figure should be removed from the paper, but that the conclusions drawn from it should be taken with a (large) pinch of salt.

4) The question of whether AC1 is present (in addition to AC5) in MSNs is central in the paper, and goes against the dogma in the field. In the current manuscript the question is tackled late in the results section (l 242) and in the discussion section, but I think it is important enough to be given in the introduction and not relegated to later sections.

MINOR POINTS

- line 30: "two types of time lags". I don't understand the point of specifying here in the abstract that *two* time lags have been considered. If you don't explain there what are those two time lags, this detail is as best confusing.

- l49: "negative PRE" should be "negative RPE"

- l71: "D1R/dopamine D2 receptors (D2R)" is cumbersome. Maybe "dopamine D1 (D1R)/ D2 (D2R) receptors" would be clearer

- l96: Fig 1C is referred to here before Fig 1A and B. Please consider correcting this.

- l123: "All higher order binding reactions (> 3) were decomposed into sets of second-order reactions. This is important because the approximation of higher order binding reactions is inappropriate to simulate temporal dynamics of molecules. Enzymatic reactions were also modeled based on the Michaelis-Menten formulation:". Actually, the Michaelis-Menten formalism is also an approximation to aggregate 3 elementary reactions and is also not very good to simulate temporal dynamics. Ideally, enzymatic reactions should also be decomposed into elementary (first- or second-order) reactions.

- l148: "D1Rs were located in the cytosolic domain": I don't see the interest of this information here? Do you refer to the experiments or the model? Since you have a single-compartment, perfectly-mixed model, it's impossible to account for different localizations anyway, so this sentence is at best confusing.

- l201: "ΔWlexp,i" should be "ΔWexp,i"

**Have all data underlying the figures and results presented in the manuscript been provided?**

Reviewer #1: Yes

Reviewer #2: Yes

PLOS authors have the option to publish the peer review history of their article (what does this mean?). If published, this will include your full peer review and any attached files.

Reviewer #1: No

Reviewer #2: No
---

## [Decision Letter · Decision Letter 2]

19 Jun 2020

Dear Dr Urakubo,

We are pleased to inform you that your manuscript 'Signaling models for dopamine-dependent temporal contiguity in striatal synaptic plasticity' has been provisionally accepted for publication in PLOS Computational Biology.

Best regards,

Joanna Jędrzejewska-Szmek, Ph.D.

Guest Editor

PLOS Computational Biology

Kim Blackwell

Deputy Editor

PLOS Computational Biology

Reviewer's Responses to Questions

**Comments to the Authors:**

Reviewer #2: The authors have satisfactorily taken into account my comments in the second revised version.

**Have all data underlying the figures and results presented in the manuscript been provided?**

Reviewer #2: No: The Matlab code of the model is provided but not the figure data.

PLOS authors have the option to publish the peer review history of their article (what does this mean?). If published, this will include your full peer review and any attached files.

Reviewer #2: Yes: Hugues Berry

---

## [Editor Report · Acceptance letter]

15 Jul 2020

PCOMPBIOL-D-19-01677R2 

Signaling models for dopamine-dependent temporal contiguity in striatal synaptic plasticity

Dear Dr Urakubo,

I am pleased to inform you that your manuscript has been formally accepted for publication in PLOS Computational Biology. Your manuscript is now with our production department and you will be notified of the publication date in due course.

With kind regards,

Sarah Hammond
